# Genome-Wide Association Study Identified a Quantitative Trait Locus and Two Candidate Genes on *Sus scrofa* Chromosome 2 Affecting Vulvar Traits of Suhuai Pigs

**DOI:** 10.3390/genes13081294

**Published:** 2022-07-22

**Authors:** Yanzhen Yin, Liming Hou, Chenxi Liu, Kaijun Li, Hao Guo, Peipei Niu, Qiang Li, Ruihua Huang, Pinghua Li

**Affiliations:** 1Institute of Swine Science, Nanjing Agricultural University, Nanjing 210095, China; 2019105013@njau.edu.cn (Y.Y.); liminghou@njau.edu.cn (L.H.); liuchenxi0018@163.com (C.L.); 2018105008@njau.edu.cn (K.L.); 2019805097@njau.edu.cn (H.G.); rhhuang@njau.edu.cn (R.H.); 2Key Laboratory in Nanjing for Evaluation and Utilization of Livestock and Poultry (Pigs) Resources, Ministry of Agriculture and Rural Areas, Nanjing Agricultural University, Nanjing 210095, China; 3Huaian Academy, Nanjing Agricultural University, Huaian 223005, China; niupeipei2@126.com; 4Huaiyin Pig Breeding Farm of Huaian City, Huaian 223322, China; 13912078083@139.com

**Keywords:** Suhuai pigs, vulvar traits, candidate genes, genome-wide association study, linkage disequilibrium and linkage analysis

## Abstract

Vulvar size and angle are meaningful traits in pig production. Sows with abnormal vulva generally show reproductive disorders. In order to excavate candidate loci and genes associated with pig’s vulvar traits, 270 Suhuai pigs with vulvar phenotype were genotyped by a porcine single nucleotide polymorphisms (SNP) Chip. Then, Chip data were imputed using resequenced data of 30 Suhuai pigs as a reference panel. Next, we estimated the heritability and performed a genome-wide association study (GWAS) for vulvar traits. The heritabilities for the traits vulvar length (VL), vulvar width (VW) and vulvar angle (VA) in this pig population were 0.23, 0.32 and 0.22, respectively. GWAS based on Chip data identified nine significant SNPs on the *Sus scrofa* chromosomes (SSC) 2, 7, 9 and 13 for VL or VW. GWAS based on imputed data identified 11 new quantitative trait loci (QTL) on SSC1, 2, 7, 8, 9, 11, 13, 16 and 17 for VL or VW. The most significant QTL for VL on SSC2 were refined to a 3.48–3.97 Mb region using linkage disequilibrium and linkage analysis (LDLA). In this refined region, *FGF19* and *CCND1*, involved in the development of the reproductive tract, cell growth and vulvar cancer, could be new candidate genes affecting VL. Our results provided potential genetic markers for the breeding of vulvar traits in pigs and deepened the understanding of the genetic mechanism of vulvar traits.

## 1. Introduction

Pork accounts for a large proportion of the meat market in China. Hence, enterprises focus on breeding pigs with high fertility. Reproductive performance is one of the most important economic traits in the pig industry since it can directly affect the population size and production efficiency of a pig farm. In breeding, vulvar size and angle reflect the development of the genital tract of a sow. The vulvar size and angle of sows should not be too small, otherwise it is difficult to carry out artificial insemination and farrowing. In recent studies, Graves et al. found that sows with larger vulva at 95–115 days had a higher proportion of estrus within 180 or 200 days than sows with smaller vulva at the same period [1]. Corredor et al. measured the vulva score categories (VSC) of 14-week-old or 15-week-old Large White pigs from three farms and found that there was a strong genetic correlation between VSC and litter size [2]. Thus, vulvar traits were very important economic traits for reproduction progress.

Most reproductive traits have complex genetic background and the traditional breeding method makes it difficult to improve these traits. In recent decades, identifying SNPs related to reproductive traits and using these associated SNP markers for marker-assisted selection or genomic selection to improve reproductive traits have become effective methods of pig breeding. However, in terms of detecting causal loci, previous studies mostly used low-density molecular markers to map QTLs, and the identified candidate loci usually covered large genomic regions with less confidence [3]. Recently, genome-wide association studies (GWAS) characterized by a high density of single nucleotide polymorphism (SNP) markers have been widely used to identify genetic mutations and QTLs for various traits, especially reproductive traits with low heritability. A large number of SNPs which are significantly related to reproductive traits, such as teat number [4], litter size [5] and sperm quality [6], have been detected. Up to date, few studies on the vulvar traits of pigs were reported and few QTLs related to the vulvar traits of pigs have been identified at the genome-wide level based on the public database of QTLdb (https://www.animalgenome.org/cgi-bin/QTLdb/SS (accessed on 1 June 2022)). Corredor et al. recently detected multiple QTLs on SSC1 2, 5, 7, 8 and 10 significantly associated with vulvar size by using GWAS in Yorkshire and Landrace populations [7]. Flossmann et al. selected sows with small vulva and normal sows in the German Landrace population for case-control analysis and found that a missense mutation on *BMP15* affected litter size of sows, but genes related to vulvar traits were not found in their study [8]. The casual genes associated with pig’s vulvar traits need to be further explored.

Suhuai pig is a synthetic pig breed, containing a 25% lineage of Chinese Huai pig and a 75% lineage of Large White pig [9]. The vulvar traits of Suhuai pig population did not undergo strong artificial selection. Therefore, there may be large variation for vulvar traits in this population, and it is an ideal population to explore key SNPs and genes associated with vulvar traits. The objectives of this study were to analyze the phenotypic variation of vulvar traits including VL, VW and VA in Suhuai pigs, to evaluate their heritability and to detect the genetic loci and genes associated with vulvar traits by performing GWAS and linkage disequilibrium and linkage analysis (LDLA).

## 2. Materials and Methods

### 2.1. Ethics Approval

All experimental protocols and details were approved by the Nanjing Agricultural University Animal Care and Use Committee (Certification No.: SYXK (Su) 2017-0007).

### 2.2. Animals and Phenotypic Collection

This study collected 270 Suhuai sows whose ages were about 160.0 ± 6.7 (Mean ± SD) days to measure the vulvar length, width and angle. All experimental individuals were raised in standard houses at the Huaiyin breeding farm (Huaian, China). Vulvar length, width and angle were measured, respectively. Vulvar length is the distance from the top to the bottom of the vulva. Vulvar width is the distance from the leftmost to the rightmost end of the widest part of the vulva. The measurement rules of vulvar length and width were consistent with that of Mills et al. [10]. Vulvar angle mainly referred to the angle between the straight line, formed by the bottom of the vulva and the bottom of the vulva fissure, and the vulva fissure (Appendix A). Iron wire was folded according to the vulvar shape and then the angle of the iron wire was measured with a protractor (Appendix A). Data from sows with estrus symptoms were excluded from the analysis. About 150 mg of ear tissue was collected from each pig and stored in a centrifuge tube filled with 75% alcohol.

### 2.3. Genotyping and Imputation

Ear tissues from 270 individuals were used to extract DNA by the phenol/chloroform method. Quantification and quality testing of genomic DNA was completed by using NanoDrop 2000 (Thermo Scientific, Waltham, MA, USA). DNA with a concentration of greater than 50 ng/μL and a value of 260/280 ranging from 1.7 to 2.1 were furthered used for SNP genotyping. For these 270 samples, 211 samples and 59 samples were genotyped using a GeneSeek GGP Porcine 80K Chip and COMPASS Porcine 50K Chip, respectively. The quality control of genotyped data was performed by using PLINK v1.09 software [11] and the code was documented in Appendix A. Samples with a call rate of SNPs > 90% were retained. SNPs with call rate > 90% and minor allele frequency (MAF) > 0.05 were retained. SNPs unmapped and on sex chromosomes were discarded. According to SNP data from 80K Chip, the missing genotypes of 50K Chip were imputed by Beagle 5.2 [12] and the code was documented in Appendix A. Imputed SNPs with MAF > 0.05 and DR² (imputed accuracy in Beagle) > 0.9 were retained. After filtering, 270 individuals and 46,622 SNPs were used for analysis. 

Meanwhile, 30 Suhuai pigs which represented the lineage of male and female pigs in the core group of Suhuai pigs were resequenced. SNP calling and filtering were executed by GATK 4.0 (Appendix A), following the previously reported process [13]. The average sequencing depth was 12 coverages, and 25,758,435 SNPs were identified. Using the resequenced data as the reference panel, the Chip data were imputed by the above method. In this imputation, 7,720,972 eligible SNPs with DR2 > 0.9 and MAF > 0.05 were retained for further analysis.

### 2.4. Descriptive Analyses and Heritability Estimation

Descriptive analyses of phenotypes including VL, VW and VA were conducted in R 3.6.1 software. The effect of environmental factors on phenotype was fitted by the general linear model in SAS 9.4 (https://odamid-apse1.oda.sas.com/SASStudio (accessed on 13 April 2022)). Significant factors were used in subsequent models. For vulvar size, birth season and measurement batch were used. For vulvar angle, only measure batch was used. Genomic data is more accurate in predicting breeding values than pedigree data [14]. In this study, we used DMU software [15] based on Chip data and phenotypical data to estimate breeding value (EBV) and calculate heritability. The formula of narrow heritability is as follows:$\;h^{2}=\sigma_{a}^{2}/\sigma_{p}^{2}$, where $\sigma_{a}^{2}$ is additive variance; $\sigma_{p}^{2}$ is phenotypic variance.

The model (Model 1) of heritability estimation contains the following effects:$$y_{ijk}=\mu +season_{i}+batch_{j}+a_{k}+e_{ijk};\;a_{k}\sim N\left(0,\;G\sigma_{a}^{2}\right);\;e_{ijk}\sim N\left(0,I\sigma_{e}^{2}\right)$$
where $y_{ijk}$ is the observed value of traits; μ is the average value of VL, VW and VA; $season_{i}$ is the effect of birth season; $batch_{j}$ is the effect of measurement batch; $a_{k}$ represents random additive effects; $e_{ijk}$ represents environmental effects; N is normal distribution; $\sigma_{e}^{2}$ is environmental variance and I is the corresponding coefficient matrix; G is the genome matrix constructed by 46622 SNPs and G is calculated by invgmatrix software [16].

### 2.5. Genome-Wide Association Study

Principal component analysis (PCA) was performed by PLINK v1.09 [11] to test population stratification. The GWAS was performed by linkage disequilibrium adjusted kinships (LDAK) software [17] to calculate the associations between genotypes and observed phenotypes (Appendix A). The mixed linear model (Model 2) of the GWAS analysis is as follows:$$Y=Xb+Wd+u+e;\;u\sim N\left(0,\;K\sigma_{a}^{2}\right);\;e\sim N\left(0,\;I\sigma_{e}^{2}\right)$$
where Y is the vector of three phenotypes; b contains birth season and measurement batch and X is incidence matrix; d is the effect of SNP and W represents the genotype matrix of SNPs; u is polygenic effect; the meanings of N, e, I, $\sigma_{e}^{2}$ and $\sigma_{a}^{2}$ are the same as model 1; K is kinship matrix calculated by pruned marker genotypes from Chip data or imputed data (pruning threshold is 0.98 and the size of window is 100 Kb) and the codes referred to Wang et al. [18]. In the mixed model, single SNP was fitted as both a fixed effect and a random effect, which would weaken the effect of a single SNP. Thus, a method named “leave one chromosome out” was used to solve these problems [19]. This method indicated that when calculating the association between one SNP and phenotype, the kinship matrix was constructed by using SNPs which were from other chromosomes. For the association results, in order to reduce the error rate, the Bonferroni method was used to correct the threshold [20]. A large number of SNPs were generated after imputation, which made the threshold set by the Bonferroni method be too strict. We pruned SNPs using “--indep-pairwise 25 5 0.8” on PLINK v1.09 [11] and then calculated the number of effective SNPs (62215) through SampleM [21]. Thus, the genome-wide threshold was equal to 0.05/N, where N is the number of effective SNPs or the number of SNPs from Chip data. In addition, SNPs with *p*-value < 1/N were considered as suggestive significant loci [22]. The CMplot package [23] was used to draw Manhattan plot and Q-Q plot (Appendix A). The former reflected the *p*-value distribution of SNPs and the latter was the comparison between the expected *p*-value and the realistic *p*-value. LD decay distance was calculated by the PopLDdecay tool [24] using 46622 SNPs (Appendix A). When the LD value (r^2^) was 0.2, the LD decay distance of 270 Suhuai pigs was about 300 Kb (Appendix A). Thus, QTLs were defined as the regions that were located in 300 Kb upstream and downstream of significant SNPs identified by GWAS.

### 2.6. Linkage Disequilibrium and Linkage Analysis (LDLA)

LDLA was conducted to refine QTLs identified by GWAS. Similar to the process of GWAS, LDLA calculated the correlation between haplotypes and phenotypes. Haplotypes were constructed through beagle 3.0 [12]. The model (Model 3) of LDLA is as follows:$$Y=Xb+Zh+e;\;h\sim N\left(0,\;K_{h}\sigma_{a}^{2}\right);\;e\sim N\left(0,\;I\sigma_{e}^{2}\right)$$
where Y, X, b, e, N, I, $\sigma_{a}^{2}$ and $\sigma_{e}^{2}$ have the same meaning as they did in model 2; h is random effect of haplotypes and Z is incidence matrix; $K_{h}$ is kinship matrix calculated by haplotypes. The confidence interval was determined when −log_10_*p* of the most significant locus decreased by 2 [25]. The code of LDLA referred to the article of Xu et al. [26] and was documented in Appendix A. The overlapping intervals mapped by GWAS and LDLA were considered as candidate QTLs in further study.

### 2.7. Phenotypic Variation Explained by SNPs

The formula $2p\left(1-p\right)b^{2}/\sigma_{y}^{2}\times 100\%$ calculated the contribution of the significant SNPs to the phenotypic variance [27]. In this formula, p is minor allele frequency of target marker; b is effect of SNP; $\sigma_{y}^{2}$ is phenotypic variance of vulvar traits.

### 2.8. Annotation of Candidate Genes

The positions of SNPs in Chip were determined according to SSC10.2 reference genome and were converted according to SSC11.1 reference genome. Genes located in intervals refined by LDLA were considered as candidate genes. Candidate genes were further searched for their function details in ensemble database (http://asia.ensembl.org/index.html (accessed on 3 May 2022)) and their biological functions were retrieved in Pubmed (https://pubmed.ncbi.nlm.nih.gov (accessed on 3 May 2022)).

## 3. Results

### 3.1. Descriptive Results of VL, VW and VA in Suhuai Pigs

There were 270 individuals with records of vulvar length and width, and 258 individuals with records of vulvar angle. Season and batch were significantly related to vulvar size and batch was significantly related to vulvar angle (Table 1). Thus, these factors would be fitted in subsequent statistical model. We summarized the characteristics of the distribution of the three traits in Suhuai pig population (Table 2, Appendix A). The ranges of VL and VW were 1.2–5.7 cm and 0.9–4.3 cm, respectively. The data showed that VL was generally greater than VW, which was consistent with vulvar characteristics of pigs in production. Their coefficients of variation were 25.46% and 27.67%, respectively, and large variations existed in the Suhuai pig population. The VA ranged from 91° to 180° and its coefficient of variation was 11.24%. When estimating heritability, only an additive effect was considered as a genetic effect. The heritabilities of three traits were 0.23, 0.32 and 0.21, respectively, suggesting these traits belonged to moderate heritability and hence, these traits were responsive to selection.

### 3.2. Imputation Description

Firstly, 50K Chip data were imputed into 80 K Chip data, and then 80K Chip data were imputed into resequenced data. After imputation, the density of SNPs within 1 Mb window was significantly improved (Appendix A). To evaluate the accuracy of imputation, 5% of the loci were randomly eliminated and imputed again. The allele concordance rate of these 5% loci was considered as the imputed accuracy. The average imputed accuracy of these two steps was 99.1% and 93.7%, respectively (Appendix A).

### 3.3. The Test of Population Stratification

PCA was executed using the “--pca 10” command in PLINK v1.09 [11]. No individual was clearly separated from the population, suggesting there was no population stratification phenomenon (Appendix A).

### 3.4. GWAS of Vulvar Traits in Suhuai Pigs

Firstly, a total of 46622 SNPs were used for preliminary GWAS analysis (Table 3, Figure 1). Among them, nine SNPs on SSC2, 7, 9 and 13 reached a chromosome significant level, four SNPs located at 3.72–3.95 Mb on SSC2 exceeded the highest threshold line and two SNPs near 3.72 Mb were significantly related to both VL and VW. There was no significant SNP for VA. Then, imputed data were further used for analysis to increase the density of markers (Table 4, Figure 1). On SSC2, 7, 9, 16 and 17, we identified six QTL regions that were significantly related to VL. The most significant SNPs (lead SNPs) on these QTL regions explained 6.71%, 5.27%, 4.81%, 4.94%, 4.55% and 3.94% of the phenotypic variance of VL, respectively. For VW, a total of five QTL regions on SSC1, 2, 8, 11 and 13 reached above the suggestive threshold line. The lead SNPs on these QTL regions explained 4.99%, 5.81%, 5.10%, 5.04% and 7.38% of the phenotypic variance of VW, respectively. Simultaneously, QTL 3.41–4.05 Mb on SSC2 were common QTL of VL and VW, which showed that this QTL contained genes or loci affecting both VL and VW. There was no significant SNP for VA. In Q-Q plots (Appendix A), the *p*-value of SNPs calculated by GWAS did not deviate obviously from the expected *p*-value. The gene inflation factors (λ) of VL, VW and VA for Chip data were 1.076, 1.077 and 1.076, respectively, and for imputed data were 1.102, 1.072 and 1.074, respectively.

### 3.5. LDLA of Vulvar Traits in Suhuai Pigs

In GWAS analysis, QTL located in 3.25–4.25 Mb on SSC2 had a considerable impact on VL. To refine this QTL interval, imputed data were used to perform the LDLA analysis of this QTL. The region of 3.48–3.97 Mb on SSC2 was identified as a candidate interval by LDLA (Figure 2). The common interval of 3.48–3.97 Mb on SSC2 identified by GWAS and LDLA was selected for further study. Genes including fibroblast growth factor 3, 4 and 19 (*FGF3*, *FGF4*, *FGF19*), LTO1 maturation factor of ABCE1 (*LTO1*) and cyclin D1 (*CCND1*) were located at this interval. After functional annotation, *FGF19* and *CCND1*, involved in the development of the reproductive tract, cell growth and vulvar cancer, were identified as the possible candidate genes of vulvar traits.

## 4. Discussion

This study measured the vulvar phenotype of 270 Suhuai pigs in total. Through descriptive analysis, the three vulvar phenotypes were obviously separated in this population, especially regarding VL and VW, which showed that this population was helpful for excavating candidate genes affecting vulvar traits. Then, these pigs were scanned by using 80K Chip (*n* = 211) or 50K Chip (*n* = 59) and 30 Suhuai pigs were resequenced using the whole genome sequencing. Two-step imputation was used to increase the marker density of data. Average imputed accuracy from 50K Chip to 80K Chip and from 80K Chip to resequenced data were 99.1% and 93.7%, respectively. Our results showed that the imputed accuracy could be related to the size of the reference population and the degree of difference in marker density between the reference population and the target population. Low density SNP Chips were directly imputed into the high-density data of the whole genome, which would affect the quality of imputation. Xu et al. imputed the 50K Chip of 246 boars into the resequenced data and obtained a genotypical accurate rate of 84.8% [26]. Yan et al. obtained a genotypical accurate rate of 89% after imputing the 60K Chips of the F3 pig population into the resequenced data [28]. Therefore, our imputed data could be used for subsequent GWAS and LDLA analysis.

In previous studies for heritability estimation, Knauer et al. recorded the vulvar width of 1225 Landrace×large white crossbred sows with an average age of 162 days, and the estimated heritability of vulvar width was 0.58 [29]. Corredor et al. estimated variance components for vulvar size in Landrace and Yorkshire gilts at 23.8 weeks of age, and showed that vulvar size had low or moderate heritability in Landrace and moderate or high heritability in Yorkshire [7]. In this study, we selected 160.0 ± 6.7-days-old Suhuai sows to explore the heritability of vulvar traits based on Chip data, and vulvar size showed moderate heritability. Genetic background, age and population size might lead to differences in heritability estimation between our study and the two previous studies mentioned above. Currently, there were no available data about the heritability of vulvar angle, and to our knowledge, we estimated the heritability of vulvar angle trait for the first time.

Based on Chip data and imputed data, several QTLs that might affect vulvar traits were mapped by GWAS. For QTLs detection, seven significant SNPs for VL or VW on SSC2, and 13 obtained by Chip data were still significant using imputed data in our study. In addition, compared to Chip data, the significant loci obtained by imputed data were more significant. The results of these two comparisons indicated that the imputed method was helpful for further excavating candidate genes. In other studies, Corredor et al. identified multiple QTLs for VL on SSC1, 2, 5, 7 and 10 and for VW on SSC1, 2 and 8 [7]. Our study also identified QTLs for VL or VW on SSC1, 2, 7 and 8, but in different regions. These differences might be due to heterogeneity among different populations. In this study, 11 significant regions on SSC1, 2, 7, 8, 9, 11, 13, 16 and 17 detected by GWAS of imputed data were all newly discovered candidate QTLs for vulvar size. However, the SNPs that were significantly related to vulvar angle were not identified, which was possibly due to a small variation coefficient within Suhuai pig population.

In addition to GWAS, LDLA was also an effective method to map and refine QTLs [26,30]. Our study used LDLA to refine the QTL region for VL on SSC2. Combining GWAS and LDLA, 3.48–3.97 Mb on SSC2 was the most likely candidate interval for VL. Meanwhile, the interval on SSC2 (3.48–3.97 Mb) was located in a QTL for teat number identified in a hybrid population between Meishan and Dutch pigs [31]. Thus, the functions of genes in this interval were annotated. Based on the functional annotation, we found that the *FGF19* belonged to the FGF family and this family was involved in the development of the female reproductive tract [32]. Furthermore, *FGF19* promoted glycogen synthesis and stimulated protein synthesis [33]. *CCND1* is a kind of cyclin coding gene in mammals and plays an important role in regulating cell division cycle and cell growth [34]. In a study of 183 patients with vulvar cancer, *CCND1* was detected to be overexpressed in vulvar cancer tissues of most patients [35]. Taken together, *FGF19* and *CCND1* on SSC2 might be two novel functional genes affecting the vulvar length. Follow-up studies are needed to excavate the functional loci of the two genes affecting vulva traits and reveal the molecular mechanism of regulating vulvar development.

## 5. Conclusions

Vulvar traits including VL, VW and VA showed moderate heritability in Suhuai pigs. A total of 11 new QTLs for VL or VW on SSC1, 2, 7, 8, 9, 11, 13, 16 and 17 were identified by GWAS. Combining GWAS and LDLA, a 3.48–3.97 Mb region on SSC2 was identified as an important QTL with a larger effect on VL, and significant SNPs in this QTL could be potential molecular markers for pig breeding. In addition, *FGF19* and *CCND1* might be new genes related to the VL of pigs. These findings will provide reference for understanding the molecular genetic basis of pig’s vulva-related traits.

## Figures and Tables

**Figure 1 genes-13-01294-f001:**
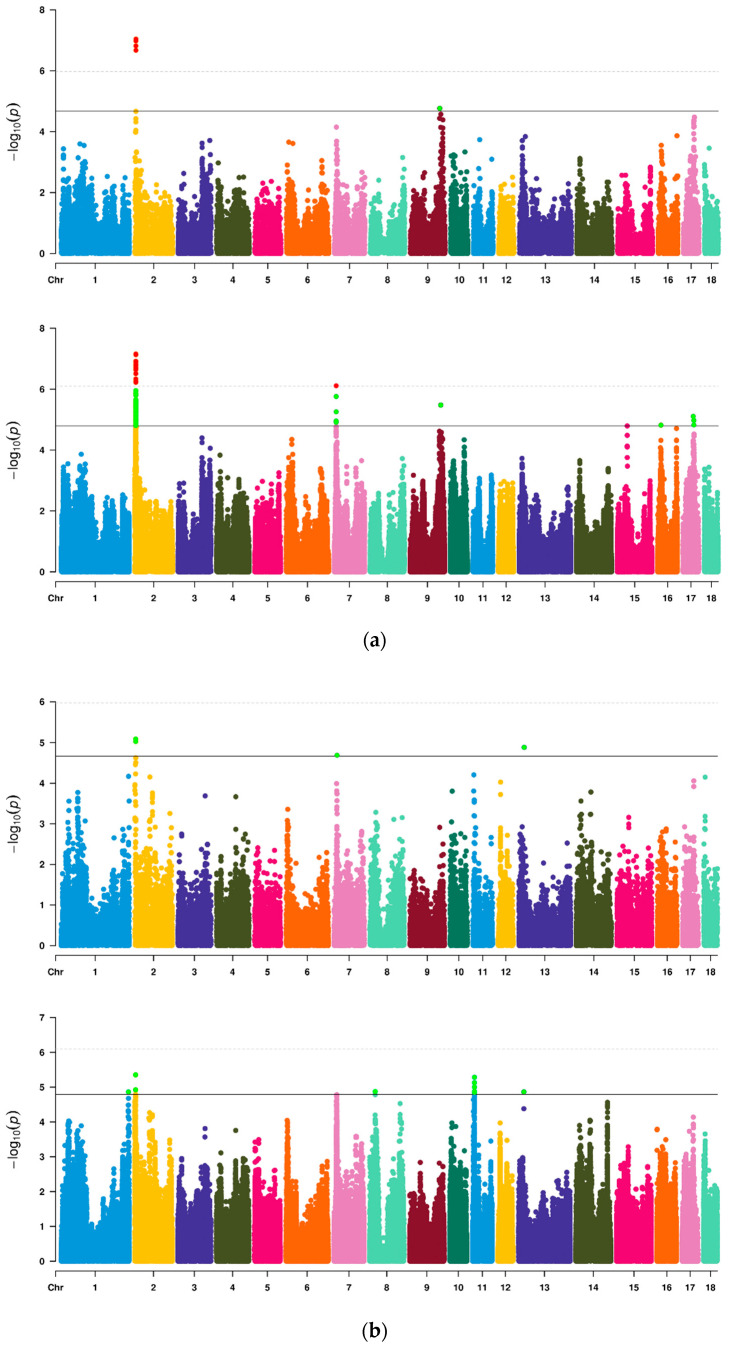
Manhattan plots of GWAS for three vulvar traits of Suhuai pigs based on Chip or imputed data. (**a**) VL; (**b**) VW; (**c**) VA. In plots (**a**–**c**), the above plot was generated using Chip data and the below plot was generated using imputed data. The solid line and the dotted line in Manhattan plots represent the suggestive threshold level and the genome wide threshold level, respectively.

**Figure 2 genes-13-01294-f002:**
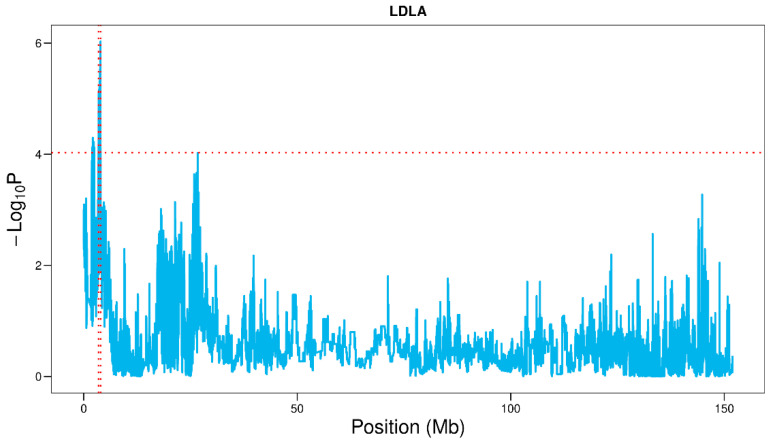
Confidence interval on SSC2 in LDLA of VL of Suhuai pigs. The two red vertical lines represent the range of the confidence interval identified by LDLA. The red horizontal line represents the threshold of LDLA.

**Table 1 genes-13-01294-t001:** Significance between environmental effects and vulvar traits of Suhuai pigs.

Traits	Number	Season	Batch
VL (cm)	270	**	**
VW (cm)	270	**	*
VA (°)	258	*	non

Note: VL is vulvar length; VW is vulvar width; VA is vulvar angle; * is significance (*p* < 0.05); ** is extreme significance (*p* < 0.01); non is nonsignificance.

**Table 2 genes-13-01294-t002:** Descriptive statistics and heritability of phenotypic data in Suhuai pigs.

Traits	Number	Max	Min	Mean	SE	CV (%)	h^2^ (SE)
VL (cm)	270	5.7	1.2	3.40	0.05	25.46	0.23 (0.13)
VW (cm)	270	4.3	0.9	2.62	0.04	27.67	0.32 (0.13)
VA (°)	258	180	91	152.42	1.07	11.24	0.21 (0.12)

Note: SE is standard error; CV is coefficient of variation; h^2^ is heritability.

**Table 3 genes-13-01294-t003:** Significant SNPs for three vulvar traits of Suhuai pigs by Chip data.

Trait	SSC	SNPs	Position	*p*-Value	Allele	MAF	PVE
VL	2	rs81344397	3,945,248	9.10 × 10^−8^	C/T	0.36	6.53%
	2	rs81323795	3,726,771	1.05 × 10^−7^	A/G	0.36	6.47%
	2	rs319327579	3,747,849	1.52 × 10^−7^	A/G	0.36	6.33%
	2	rs336379732	3,831,753	2.13 × 10^−7^	T/C	0.36	6.22%
	9	rs326835497	118,063,093	1.71 × 10^−5^	A/G	0.27	5.09%
VW	2	rs81323795	3,726,771	8.13 × 10^−6^	A/G	0.36	5.45%
	2	rs319327579	3,747,849	9.39 × 10^−6^	A/G	0.36	5.36%
	7	rs80898557	11,989,354	2.04 × 10^−5^	T/C	0.43	4.33%
	13	rs320033947	21,085,416	1.31 × 10^−5^	G/T	0.43	8.19%

Note: MAF is the minor allele frequency; PVE represents the proportion of phenotypical variance explained by SNPs. The physical positions were annotated by *Sus scrofa* 11.1 reference genome.

**Table 4 genes-13-01294-t004:** Candidate QTLs for three vulvar traits of Suhuai pigs by imputed data.

Traits	SSC	QTL Region (Mb)	Position of Lead SNP (bp)	*p*-Value of Lead SNP	PVE
VL	2	3.25–4.25	3,833,112	6.98 × 10^−8^	6.71%
	7	7.23–8.07	7,763,002	7.74 × 10^−7^	5.27%
	9	123.25–123.87	123,550,278	3.33 × 10^−6^	4.81%
	16	12.66–13.26	12,961,112	1.52 × 10^−5^	4.94%
	17	40.60–41.20	40,902,044	7.82 × 10^−6^	4.55%
	17	43.10–43.72	43,404,836	1.06 × 10^−5^	3.94%
VW	1	271.53–272.13	271,827,161	1.37 × 10^−5^	4.99%
	2	3.41–4.05	3,726,718	4.40 × 10^−6^	5.81%
	8	22.30–22.93	22,602,662	1.34 × 10^−5^	5.10%
	11	7.29–8.10	7,650,870	5.19 × 10^−6^	5.04%
	13	20.78–21.38	21,080,852	1.37 × 10^−5^	7.38%

Note: Lead SNP is the most significant SNP in one region. The physical positions were annotated by *Sus scrofa* 11.1 reference genome.

## Data Availability

The Chip data and phenotype data of the 270 Suhuai pigs were deposited at the figshare repository (https://doi.org/10.6084/m9.figshare.19494398.v1 (accessed on 13 April 2022)). The resequenced data of 30 Suhuai pigs were deposited at NCBI (PRJNA791712).

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
