# Peer review of "Genome-Wide Association Study Identified a Quantitative Trait Locus and Two Candidate Genes on Sus scrofa Chromosome 2 Affecting Vulvar Traits of Suhuai Pigs"

_genes, 2022, doi:10.3390/genes13081294_

Round 1

Reviewer 1 Report

It is well known that oestrus status affects vulvar forms and dimensions. Authors say nothing, if I am not wrong, on the oestrus status for the measurement of phenotypes. Probably, nobody considered the importance of this aspect. The consequence is the affordability of phenotypes used throughout the paper. Please, consider that at 160 days of age sows should have started a normal reproductive activity.

Ask a doctor in veterinary medicine the effect of the heat on the vulva of a female mammal. You will derive the consequences on the phenotypes measured in the paper.

Author Response

Response to Reviewer 1 Comments

Dear Editor,

First of all, thanks for your kind comments and valuable suggestions on our manuscript. We have revised our manuscript according to your comments.

Point 1: It is well known that oestrus status affects vulvar forms and dimensions. Authors say nothing, if I am not wrong, on the oestrus status for the measurement of phenotypes. Probably, nobody considered the importance of this aspect. The consequence is the affordability of phenotypes used throughout the paper. Please, consider that at 160 days of age sows should have started a normal reproductive activity.

Response 1: Thanks for your valuable comment and remind. First, SuHuai pigs generally oestrus after 180 days of age. Then, we will observe whether the sow has oestrus symptoms (fever, vulva swelling, etc.) during the determination. If there are the above phenomena, this individual will not be determined. Because estrus rarely occurs during the determination, it is not described in detail in the article. We added a description of this detail in Section 2.2 (animals and phenotypic collection) in the revised article. (line 91-94)

Point 2: Ask a doctor in veterinary medicine the effect of the heat on the vulva of a female mammal. You will derive the consequences on the phenotypes measured in the paper.

Response 2: Thanks for your valuable comment. Estrus status can be an important factor affecting the vulvar phenotype of sows. We describe this detail in Response 1.

Meanwhile, we have improved the sentence structure and readability of the manuscript. Thank you again for the kind advices. If you have any questions about this revision, please don’t hesitate to let me know.

Yours sincerely,

Dr Pinghua Li

Reviewer 2 Report

- In line 19, replace for "by porcine";

- In line 22, "The heritability of vulvar length (VL), vulvar width 22 (VW) ... ", rewrite: "The heritability for the traits vulvar length... in this pig population were.... , respectively."

- In line 93, " phenol/chloroform" add " phenol/chloroform method".

- In line 100, replace "Individuals with call rate..." with"Samples with call rate..."

- In item "2.4 Descriptive analysis and heritability estimation", what is the batch effect in the statistical model?

- In 123 line, remove "â„Ž 2 is narrow heritability"

- In 164 line, add in 2.6 item the abreviation "LDLA" after the analysis description

- In line 200, add "and hence these traits are responsive to selection".

Author Response

Response to Reviewer 2 Comments

Dear Editor,

First of all, thanks for your kind comments and valuable suggestions on our manuscript. We have revised our manuscript according to your comments.

Point 1: In line 19, replace for "by porcine".

Response 1: Thanks for your suggestion. We have revised it according to your comment, thank you (line 19).

Point 2: In line 22, "The heritability of vulvar length (VL), vulvar width 22 (VW) ... ", rewrite: "The heritability for the traits vulvar length... in this pig population were.... , respectively.

Response 2: Thanks for your suggestion. We have revised it according to your comment, thank you (line 22).

Point 3: In line 93, " phenol/chloroform" add " phenol/chloroform method".

Response 3: Thanks for your suggestion. We have revised it according to your comment, thank you (line 97).

Point 4: In line 100, replace "Individuals with call rate..." with"Samples with call rate...".

Response 4: Thanks for your suggestion. We have revised it according to your comment, thank you (line 104).

Point 5: In item "2.4 Descriptive analysis and heritability estimation", what is the batch effect in the statistical model?

Response 5: Thanks for your valuable comment. We collected the vulvar phenotype in multiple batches, so batch was added to the statistical model to correct for the effect of different batch.

Point 6: In 123 line, remove "â„Ž 2 is narrow heritability".

Response 6: Thanks for your suggestion. We have revised it according to your comment, thank you (line 127).

Point 7: In 164 line, add in 2.6 item the abreviation "LDLA" after the analysis description.

Response 7: Thanks for your suggestion. We have revised it according to your comment, thank you (line 168).

Point 8: In line 200, add "and hence these traits are responsive to selection".

Response 8: Thanks for your suggestion. We have revised it according to your comment, thank you (line 204).

Meanwhile, we have improved the sentence structure and readability of the manuscript in attachment. Thank you again for the kind advices. If you have any questions about this revision, please don’t hesitate to let me know.

Yours sincerely,

Dr Pinghua Li

Round 2

Reviewer 1 Report

"Considering the effect of oestrus on vulvar phenotype, we 91 will observe whether the sow has oestrus symptoms (fever, vulva swelling, etc.) during  the determination. If there are the above phenomena, this individual will be eliminated. "

Data from sows with oestrus symptoms were excluded from the analysis. Maybe, this way is better.

Author Response

Response to Reviewer 1 Comments

Dear Editor,

First of all, thanks for your kind comments and valuable suggestions on our manuscript. We have revised our manuscript according to your comments.

Point 1: "Considering the effect of oestrus on vulvar phenotype, we 91 will observe whether the sow has oestrus symptoms (fever, vulva swelling, etc.) during  the determination. If there are the above phenomena, this individual will be eliminated."

Data from sows with oestrus symptoms were excluded from the analysis. Maybe, this way is better..

Response 1: Thanks for your suggestion. We have revised it according to your comment, thank you (line 91).

Meanwhile, we have improved the sentence structure and readability of the manuscript. Thank you again for the kind advices. If you have any questions about this revision, please don’t hesitate to let me know.

Yours sincerely,

Dr Pinghua Li